# KRAS Mutations in Solid Tumors: Characteristics, Current Therapeutic Strategy, and Potential Treatment Exploration

**DOI:** 10.3390/jcm12020709

**Published:** 2023-01-16

**Authors:** Yunkai Yang, Huan Zhang, Shanshan Huang, Qian Chu

**Affiliations:** Department of Oncology, Tongji Hospital, Tongji Medical College, Huazhong University of Science and Technology, Wuhan 430030, China

**Keywords:** KRAS mutations, therapeutic strategy, drug resistance, oncogenic mechanisms, immune microenvironment

## Abstract

Kristen rat sarcoma (KRAS) gene is one of the most common mutated oncogenes in solid tumors. Yet, KRAS inhibitors did not follow suit with the development of targeted therapy, for the structure of KRAS has been considered as being implausible to target for decades. Chemotherapy was the initial recommended therapy for KRAS-mutant cancer patients, which was then replaced by or combined with immunotherapy. KRAS G12C inhibitors became the most recent breakthrough in targeted therapy, with Sotorasib being approved by the Food and Drug Administration (FDA) based on its significant efficacy in multiple clinical studies. However, the subtypes of the KRAS mutations are complex, and the development of inhibitors targeting non-G12C subtypes is still at a relatively early stage. In addition, the monotherapy of KRAS inhibitors has accumulated possible resistance, acquiring the exploration of combination therapies or next-generation KRAS inhibitors. Thus, other non-target, conventional therapies have also been considered as being promising. Here in this review, we went through the characteristics of KRAS mutations in cancer patients, and the prognostic effect that it poses on different therapies and advanced therapeutic strategy, as well as cutting-edge research on the mechanisms of drug resistance, tumor development, and the immune microenvironment.

## 1. Introduction

Kirsten rat sarcoma (KRAS) gene is one of the most common mutated oncogenes in numerous cancer types, such as non-small cell lung cancer (NSCLC), colorectal cancer (CRC), and pancreatic ductal adenocarcinoma (PDAC) [1]. It is a member of the rat sarcoma (RAS) viral oncogene family, and was first found in 1982 on the short arm of chromosome 12 in human lung cancer cells [2]. With KRAS lacking classical drug binding sites, the development of an inhibitor targeting KRAS mutations is extremely challenging, resulting in no breakthrough in KRAS-targeted therapy for a long time. Traditionally, chemotherapy has been used for patients with KRAS-mutant lung cancer and other solid tumors. With the development of immunotherapy in recent years, strategies have also been gradually developed for the combination of immunotherapy and chemotherapy for patients with KRAS mutations. The year of 2021 saw a breakthrough in KRAS-targeted drugs, with Sotorasib being approved for clinical use in patients with NSCLC and other solid tumors with KRAS G12C mutations [3]. Adagrasib, another KRAS G12C inhibitor, also showed promising results in a phase-2 clinical trial of NSCLC patients in 2022 [4]. As research targeting the immune microenvironment of tumors has flourished, KRAS-related studies have also emerged. This review intends to summarize the characteristics of KRAS mutations in solid tumors, update the latest clinical therapeutic strategies, and discuss the mechanisms of drug resistance and tumor development, as well as the immune microenvironment.

## 2. Molecular Biological Functions of KRAS

The protein that KRAS encodes is a membrane-bound regulatory protein (G protein) that acts by binding guanosine triphosphatases (GTPases) [5]. It usually acts as the switch altering its GTP-bound active state and GDP-bound inactive state, with a resynthesis half-life of ~24 h [6,7,8,9]. Guanine nucleotide exchange factor (GEF), such as son of sevenless isoform 1 (SOS1) protein, functions to promote the active form of KRAS during this switch [10,11]. Such an alteration causes conformational changes of KRAS binding Raf proteins, activating downstream effectors that are in charge of cellular growth, differentiation, and survival. The deactivation of GTPase caused by oncogenic mutation accumulates the KRAS-GDP state, initiating downstream pathways such as mitogen-activated protein kinase (MAPK) pathway and phosphoinositide 3-kinase (PI3K) pathway) [12,13]. In the MAPK pathway, activated KRAS-GTP causes a rapid increase in the number of serine/threonine-specific protein kinase (RAF) within cells localizing to the plasma membrane, leading to conformational change. The then activated RAF further binds to mitogen-activated protein kinase 1/2 (MEK1/2), activating extracellular regulated protein kinases 1/2 (ERK1/2) via phosphorylation [14]. In the PI3K pathway, KRAS GTP binds to the p110s site of PI3K, leading to PI3K activation, which converts the phosphatidylinositol 4, 5-diphos-phate (PIP2) to phosphatidylinositol 3,4,5-triphosphate (PIP3). PIP3 promotes the phosphorylation of serine/threonine-protein kinase (AKT) by phosphoinositide-dependent kinase 1 (PDK1), activating the mammalian target of rapamycin (mTOR) pathway [15]. Activated downstream signaling pathways regulate cell proliferation, differentiation, migration, and other cellular life activities [14,16].

## 3. KRAS Mutations in Cancers

### 3.1. Frequencies and Types of KRAS Mutations

According to The Cancer Genome Atlas (TCGA) database, KRAS mutations are present in approximately 11.6% of all carcinomas, the mutation rates and subtypes of which vary widely between tumors. KRAS mutations are most common in PDAC, with 81.72% of patients presenting KRAS mutations. CRC had the second highest KRAS mutation frequency, with a mutation frequency of 37.97%. KRAS mutations are present in approximately 21.20% of patients with NSCLC. In addition, KRAS mutations are mainly found in choloangiocarcinoma, uterine endometrial carcinoma, testicular germ cell cancer, and cervical squamous cell carcinoma, with a mutation frequency of about 12.7%, 14.1%, 11.7%, and 4.3%, respectively. KRAS mutations mainly contain 21 missense mutations, with G12D (29.19%), G12V (22.97%), and G12C (13.43%) being the most common. Studies from the TCGA database suggest that the prevalence of KRAS mutations in PDAC is the highest (over 80%) of all gene mutations, with G12D being the most common subtype; and that in CRC, KRAS G12D, and G12V are the two most common mutant subtypes. In all lung cancers, KRAS mutations mainly accounted for 11.2–25.3% of all mutations, with the KRAS G12C-type mutations accounting for 2.8–15% [17]. Specifically in NSCLC, KRAS G12C is the most common mutant subtype, accounting for roughly 45% of all KRAS mutations, followed by G12V and G12D [18]. Apart from the common tumor types, the landscape of the KRAS mutation in rare tumors is investigated, with an overall mutation rate of 8.7%; and G12D and G12V, along with G13D, are the most common subtypes [19]. The KRAS mutation frequency and subtype proportion in common cancers are summarized in Table 1. Mutations in KRAS, different mutant subtypes of KRAS, and other genetic co-alterations with the KRAS mutation may all have an impact on the clinicopathological features and prognoses of cancer patients.

### 3.2. Clinicopathological Characteristics of KRAS Mutations

KRAS mutations are associated with specific clinicopathological features in different tumors. In CRC, A study of left-sided, microsatellite stable CRC found that the proportion of KRAS-mutant patients is higher in the lung-metastatic cohort, while it is lower in the liver-metastatic cohort [20]. Another study of metastatic CRC showed that the regression of the KRAS mutation was associated with a better prognosis and oligo-metastatic status [21]. In NSCLC, KRAS mutations are seen in about 30% of lung adenocarcinomas and 5% of squamous lung cancers, in 26% of Westerners and 11% of Asians, and also in 30% of smokers and 10% of nonsmokers [22]. A study of lung adenocarcinoma patients found that KRAS mutations were significantly associated with older age (>45 years old) at diagnosis [23]. In addition, liver metastases and brain metastases occurred more frequently in NSCLC patients with KRAS mutations than in wild-type [24]. KRAS mutation was also found to be a biomarker of lower differentiation in neuroendocrine tumors (NETs).

### 3.3. Prognostic Value of KRAS Mutations

Numerous studies have shown that KRAS mutations can have an impact on the prognoses of cancer patients, although the results of different studies are somewhat contradictory regarding their specific impacts. Most of the current studies show an association with poor prognosis, and that subtypes and co-mutations may also have an impact.

#### 3.3.1. Overall Impact of KRAS Mutations on Prognosis

In PDAC, more studies tend to consider KRAS mutations as a predictor of poor prognosis. A study including 39 PDAC patients found that patients with KRAS mutations, specifically the subtype of G12D, had a significantly worse overall survival (OS) and disease-free survival (DFS) [25]. A study based on 110 PDAC patients reported that those with KRAS mutations have significantly shorter PFS and OS (5.3/6.9 months (*p* = 0.044) vs. 11.8/19.9 months (*p* = 0.037), respectively) [26]. Another study detecting the circulating cell-free tumor-DNA (cft-DNA) of 29 PDAC patients showed that the survival of patients with detectable plasma KRAS mutations pre-treatment was significantly worse (16.8 months vs. not reached, *p* < 0.031) [27]. A study comparing PDAC patients treated with surgery to chemotherapy showed that the detectable plasma KRAS mutation is an independent predictor of early recurrence after surgery, while not showing a significant difference in PFS after chemotherapy [28].

KRAS mutations have also been reported to have a negative impact on patients with colorectal cancer. A study of hepatic arterial infusion (HAI) pump therapy in unresectable colorectal liver metastases showed that patients with KRAS mutations had worse responses to HAI chemotherapy, compared with wild-type patients [29]. A study of epidermal growth factor receptor (EGFR) inhibitors combined with third-line chemotherapy in metastatic CRC patients found that the RAS mutation was an independent predictor for shorter PFS [30]. A retrospective study based on locally advanced rectal cancer patients treated with neoadjuvant chemoradiation therapy (nCRT) and proctectomy showed that KRAS mutations were not associated with a pathologic complete response (pCR), yet they were independently related to a worse prognosis [31]. Another study of stage IV CRC patients found a shorter OS for patients with KRAS mutations versus wild-type status (*p* = 0.004) [32]. A study using the National Cancer Database reported that KRAS mutations presented a worse OS than the KRAS wild-type in metastatic CRC patients under the age 70, while no significant association was detected above the age of 70 [33].

In the study of NSCLC, the prognostic impact of KRAS mutations is controversial. A retrospective study of Durvalumab in patients with unresectable stage III NSCLC showed that patients with KRAS mutations had a better median PFS than EGFR and BRAF genetic mutations (not reached vs. 8.1 months vs. 7.8 months, *p* = 0.02) [34]. A study of immune checkpoint inhibitors (ICIs) in lung cancer patients with brain metastasis showed that KRAS mutations may drive a better efficacy of immunotherapy [35]. Despite these optimistic results, some reported that KRAS mutations might not have a significant impact on the prognoses of NSCLC patients. A multicenter cohort study of 1017 lung cancer patients of immunotherapy showed that the KRAS mutations had no significant impact on the response to ICIs in NSCLC patients [36]. Another meta-analysis regarding NSCLC patients concurred that no statistical OS improvement in KRAS mutant or KRAS wild-type patients [37]. A study of abemaciclib (a selective small-molecule CDK 4/6 inhibitor) combined with pembrolizumab in stage IV NSCLC patients showed that KRAS-mutant patients presented a better efficacy, yet a greater toxicity [38]. A study of stage IV lung adenocarcinoma patients treated with pembrolizumab as first-line monotherapy showed that KRAS has no prognostic impact on such treatment [39]. Another study of ICIs treating NSCLC patients with high programmed death ligand 1 (PD-L1) expression showed that the KRAS mutation status had no significant impact on the efficacy or safety of ICIs, though with a non-significant trend of worse survival in patients with KRAS G12C mutations [40]. A real-world study with 150 lung adenocarcinoma patients reported no significant difference in the PFS, OS, as well as first-line chemotherapy response among patients with or without KRAS mutations [41]. In conclusion, KRAS mutations may not have a significant impact on the prognosis of NSCLC patients.

#### 3.3.2. Impacts of Different KRAS Mutation Subtypes on Prognosis

Different KRAS mutant subtypes may also present different impacts on prognosis. In NSCLC, a single-center cohort study of non-squamous NSCLC patients with KRAS mutations reported that patients with KRAS G12C mutations had a higher response rate (53.8% versus 8.3%, *p* = 0.030) and a longer PFS (4.8 months versus 2.1 months, *p* = 0.028) than those with mutations of other KRAS subtypes [42]. Such a finding might be controversial, considering that another real-world study based on 1039 NSCLC patients found no significant difference in treatment with ICI between patients with the KRAS wild type, G12C mutant, and other KRAS subtype mutations [43]. In CRC, a retrospective study comparing KRAS G12C mutations to other KRAS mutations in metastatic CRC patients treated with first-line chemotherapy plus bevacizumab showed that KRAS G12C had a significantly worse response rate (RR) than other subtypes (*p* = 0.017), while no difference in PFS (*p* = 0.76) and OS (*p* = 0.56) was observed [44]. A study of 419 CRC patients with unresectable liver metastases showed that the KRAS A146 mutations had a high tumor burden (TMB) and a worse OS compared with the G12 subtypes (median OS 10.7 months vs. 26.4 months; *p* = 0.003) [45].

#### 3.3.3. Impact of KRAS Co-Alterations on Prognosis

The presence of genetic co-mutations in patients with KRAS mutations has also been reported in studies. According to current studies related to lung cancer patients with KRAS mutations, the most common co-mutated gene is tumor protein p53 gene (TP53), which accounts for about 39–42% of patients with KRAS mutations, followed by serine/threonine kinase 11 gene (STK11), accounting for about 20–30%, and kelch-like ECH associated protein 1 gene (KEAP1), accounting for about 13–27% [46,47]. In addition, there are co-mutated genes such as ATM serine/threonine kinase gene (ATM), MNNG HOS Transforming gene (MET), and erb-b2 receptor tyrosine kinase 2 gene (ERBB2), which account for more than 10% of patients with KRAS mutated lung cancer [46].

Several features that coexist with KRAS mutations also have an impact on the prognosis of patients. In a retrospective study of 587 resected PDAC patients, the co-mutation of TP53 and KRAS G12D is shown to be an independent predictor of better OS and recurrence-free survival (RFS) [48]. In a study of HER2-mutant advanced gastric cancer patients treated with trastuzumab, KRAS mutation was found to be a predictor of insufficient efficacy and poor prognosis [49]. As for NSCLC, a study of 50 EGFR T790M-mutant NSCLC patients found that patients with TP53 R237C or KRAS G12V mutations cannot benefit from subsequent osimertinib treatment [50]. Another study of 946 patients reported that concomitant KRAS mutation and copy number gain (copy number ≥2) was a predictor of worse survival [51]. KRAS/STK11 co-mutations was reported to have a worse survival among all KRAS-mutant metastatic lung adenocarcinoma patients [18]. In a study of NSCLC patients treated with immunotherapy, patients with TP53/KRAS co-mutation were found to have a significantly longer PFS (5.8 vs. 2.6 months, *p* = 0.005) [52]. Another study of advanced lung adenocarcinoma showed that patients with KRAS mutations of all gene mutations had the best response to immunotherapy [53]. However, co-mutations may alter such efficacy, such as KRAS-mutant patients with STK11 or KEAP1, as co-mutations have a poorer prognosis for immunotherapy than KRAS wild-type lung adenocarcinoma patients [54].

Except from co-mutations, a High src homology region 2 domain-containing phosphatase 2 (SHP2) expression was reported to be a predictor of a better survival rate and a better efficacy of immunotherapy in a study of 61 KRAS-mutant advanced NSCLC patients [55]. A low expression level of serum deprivation protein response (SDPR) was found to be suppressing the immune system, independently correlated with a shorter OS in KRAS-mutant NSCLC patients [56]. Furthermore, a study based on 25 PDAC patients after R0/R1-resection showed that KRAS mutations combining a high carbohydrate antigen 19-9 (CA 19-9) level is a better predictor than individual markers, with an impact on early relapse and poorer OS, compared to those with KRAS wild-type or a low CA 19-9 level [57]. A high expression of both tyrosine phosphatase PTPN2 and LAMA3/AC245041.2 was found to be significantly related to the poor prognosis of KRAS-mutant patients with PDAC [58,59].

## 4. Therapeutic Strategies in KRAS-Mutant Cancers

The high incidence of KRAS makes it one of the most attractive and challenging therapeutic targets. Traditionally, patients with KRAS-mutant solid tumors were treated with chemotherapy, and the standard of care for patients with KRAS-mutant solid tumors is chemotherapy. In CRC, the median PFS of KRAS-mutant patients is 11.6 months, significantly worse than patients with wild-type KRAS [60]. Albuminpaclitaxel combined with gemcitabine is a first-line chemotherapy regimen that has been widely applied to pancreatic cancer, but most patients rapidly develop drug resistance after several courses of treatment [61]. The average OS of NSCLC patients with KRAS mutations treated with chemotherapy is less than 2 years [22]. The efficacy of conventionally applied chemotherapy for patients with KRAS mutations is limited and needs to be improved.

Since the development of immunotherapy, treatment options for patients with KRAS mutations have gradually entered the era of immunotherapy. In addition, many studies have explored the indirectly targeted therapy of upstream and downstream KRAS pathways, and recently, there has been a breakthrough in the directly targeted therapy of KRAS. The immunotherapeutic agents and targeted therapeutic agents currently under research, and their effects on the upstream and downstream pathways of KRAS can be seen in Figure 1.

### 4.1. Immunotherapy in KRAS-Mutant Cancers

With the application of monoclonal antibodies targeting programmed death 1 (PD-1) and its primary ligand PD-L1, the treatment paradigm for most advanced solid tumors has been fundamentally altered. Further studies have also been conducted in KRAS mutant tumors, and data shows that KRAS mutations may reshape the tumor immune microenvironment [62]. An analyzer powered by artificial intelligence based on hematoxylin and eosin showed that KRAS mutations were mostly found in the inflamed subtype of the immune microenvironment [63]. A study based on 202 patients of lung adenocarcinoma found that tumor-infiltrating lymphocytes (TILs) were relatively abundant in more than 60% of cases, TTF1 positivity was found in 78.7% of cases, and PD-L1 positivity was found in 25.2% of cases [64]. The mechanisms of KRAS mutations regulating the immune microenvironment may include the secretion of neutrophil chemokines, the downregulation of major histocompatibility complex I (MHC I), the induction of regulatory T (Treg) cells, and the upregulation of PD-L1 [65]. These results all suggest that patients with KRAS mutations may benefit from immunotherapy.

In recent years, immunotherapy has shown advantages in treating patients with KRAS mutations, especially in NSCLC. A single-center, retrospective cohort study using ICIs as a first-line treatment for KRAS-mutant advanced NSCLC patients reported a median PFS of 16.2 months and a median OS of 31.3 months [66]. A meta-analysis of randomized controlled trials (RCTs) comparing anti-PD-(L)1 with chemo-monotherapy for advanced KRAS-mutant NSCLC showed that patients treated with first- or second-line anti-PD-(L)1 with or without chemotherapy had longer OS and PFS than chemotherapy alone [67]. A study based on 44 NSCLC patients with high PD-L1 expression showed that patients with KRAS G12C mutations have a significantly longer PFS when treated with anti-PD-1 immunotherapy [68]. A meta-analysis discussing the efficacy of immunotherapy in NSCLC patients with genetic mutations showed that ICIs significantly prolonged the OS of patients with KRAS mutations [69].

Combination strategies of immunotherapy and chemotherapy are also emerging, with most of the studies supporting that combination therapy has a better efficacy than monotherapy. A real-world retrospective study of 497 KRAS-mutant NSCLC patients reported that patients had a significantly longer survival when treated with chemoimmunotherapy than immunotherapy alone (median PFS 13.9 vs. 5.2 months, *p* = 0.049) [70]. According to the phase 3 Impower 150 trial, atezolizumab plus bevacizumab and chemotherapy is an effective first-line therapy for KRAS-mutant NSCLC patients with STK11, KEAP1, or TP53 co-mutations [71]. A retrospective single-center study comparing PD-1 inhibitors combined with nab-paclitaxel plus gemcitabine (AG) chemotherapy versus AG as the first-line treatment of advanced pancreatic cancer reported that PD-1+AG could improve the OS of patients with KRAS/TP53 co-mutations [72]. However, there also have been studies reporting different findings. A cohort study comparing ICIs monotherapy with chemoimmunotherapy for NSCLC patients showed a better rate of survival in KRAS-mutant patients than wild-type patients, yet no significant difference was found between treatment strategies in the subgroup of KRAS-mutant patients [73].

In summary, immunotherapy combined with chemotherapy appears to provide greater clinical benefits in patients with KRAS mutations compared to immune monotherapy. However, it is worth noting that the PD-L1 expression level plays a more dominant role than KRAS mutations.

### 4.2. Direct and Indirect Inhibitors of KRAS

#### 4.2.1. Directly Targeted Therapy

Direct targeting KRAS has long been considered difficult. As researchers have learned more about the structure of KRAS and the complex interactions involved in the RAS signaling protein family, they have been able to break through the perception that KRAS targets are “undruggable”, developing direct inhibitors of KRAS targets in recent years [74]. Drugs directly targeting KRAS have recently emerged with promising results. Sotorasib (AMG510) is currently the only FDA-approved tyrosine kinase inhibitor (TKI) targeting KRAS G12C [3]. In the CodeBreaK 100 phase 2 single-arm trial, Sotorasib had an ORR of 37% and a median PFS of 6.7 months [75]. Adagrasib (MRTX849) performs well in the KRYSTAL-1 phase 1 and phase 2 trials, and it is expected to be the next approved drug targeting KRAS G12C [4,76]. ASP2453, a novel KRAS G12C inhibitor, showed antitumor efficacy in the preclinical models of KRAS G12C mutant cancers [77]. The latest clinical studies of all KRAS inhibitors are presented in Table 2.

Current clinical studies of KRAS inhibitors have shown promising results with multiple agents in a variety of solid tumors. The structural features of each subtype make G12C a breakthrough for KRAS inhibitors, but they also make it difficult for other subtypes to benefit from KRAS G12C inhibitors. Inhibitors for other subtypes are under investigation and are to be expected. However, most patients have previously received systemic immunotherapy or chemotherapy, so the evidence for their use as first-line therapy remains to be explored. Treatment strategies in combination with other therapies are still under investigation, yet the results are worth expecting.

#### 4.2.2. Indirectly Targeted Therapy

Apart from drugs directly targeting KRAS, a number of indirectly targeted therapies targeting its upstream or downstream signaling pathway have also been developed. A study found that AZD0424 (an SRC inhibitor), when combined with MEK inhibitors (such as trametinib), inhibits tumor growth more than MEK inhibitor monotherapy, but does not reverse pre-existing MEK inhibitor resistance [78]. A study of KRAS G12R-mutant pancreatic cancer patients treated with selumetinib (KOSELUGO™; ARRY-142886, an oral MEK1/2 inhibitor) showed a median PFS of 3.0 months and a median OS of 9 months. Expectations were not met, and this group of patients should be considered for combination with other therapies [79]. The Traf2- and Nck-interacting protein kinase (TNIK) inhibitor NCB-0846 was found to enhance cell death induced by the BCL-X(L) inhibitor ABT-263 in KRAS/BRAF mutant cells, which may be a new combination treatment strategy for the KRAS/BRAF-mutant CRC [80]. Crenolanib, a TKI targeting tyrosine kinase receptors such as platelet-derived growth factor receptor A (PDGFRA), platelet-derived growth factor receptor B (PDGFRB), and FMS-like tyrosine kinase-3 (FLT3), may have clinical benefit for KRAS/BRAF-mutant CRC patients [81]. 

Some combination strategies have shown benefits as well. There is a retrospective study comparing bevacizumab plus capecitabine, with capecitabine monotherapy for KRAS-mutant metastatic CRC showing that combination therapy was better tolerated, and also contributing a longer PFS (9.0 months vs. 7.2 months, *p* < 0.05) [82]. Another study found that conventional mFOLFOX6 chemotherapy combined with cetuximab for KRAS-mutant CRC patients showed was shown to improve efficacy, reduce the overall incidence of adverse events (AEs), improve OS, and extend overall patient survival when adding simvastatin [83]. A phase 1 study of binimetinib (MEK inhibitor) plus carboplatin and pemetrexed chemotherapy for stage-IV non-squamous NSCLC showed that the ORR of patients with KRAS/NRAS mutations was 62.5%, while wild-type patients had an ORR of 25% [84]. A phase 1b, multi-center study of binimetinib plus pemetrexed and cisplatin chemotherapy, followed by the maintenance of binimetinib and pemetrexed for advanced KRAS-mutant NSCLC, reported an ORR of 33%, a median PFS of 5.7 months, and a median OS of 6.5 months, with no unacceptable AEs [85]. A study found that the sequential triple therapy of anti-PD-1 ICIs sequentially after fulvestrant (anti-estrogen) plus dacomitinib (a pan-HER inhibitor) had the potential for treating KRAS-mutant lung cancer [86]. In pancreatic cancer, an open-label, phase 2 RCT comparing stereotactic body radiotherapy (SBRT) plus pembrolizumab and trametinib versus SBRT plus gemcitabine for locally recurrent pancreatic cancer post-operation reported that patients with KRAS mutations as well as high PD-L1 expression had a median OS of 14.9 and 12.8 months, respectively, with the major AEs being increased blood bilirubin and liver impairment [87]. The most recent clinical trials of non-KRAS-targeted therapies are presented in Table 3.

### 4.3. Other Unconventional Therapies

Some new progress has been made to cope with arising resistance towards conventional therapies. In CRC, a mutant Hydra actinoporin-like-toxin-1 (mHALT-1) immunotoxin was developed to treat KRAS G12V-mutant CRC patients, which showed cytotoxic effects in KRAS-mutant CRC cells [88]. 2-methoxyestradiol (2-ME), the superoxide dismutase inhibitor, showed synergistic effects with ABT-263 (a BCL-X(L) targeting agent) in KRAS-mutant CRC cell lines [89]. A study found that exosomes loaded with clustered regularly interspaced palindromic repeats (CRISPR)/Cas9 could target the KRAS G12D-mutant allele in PDAC cells to suppress cell proliferation and tumor growth, making it a promising treatment strategy [90]. M-LIP-CLT, a hybrid nanoplatform capable of fusing Celastrol (CLT)-Loaded PEGylated lipids with the DC2.4 cell membrane, is an effective drug delivery system for PDAC-targeted therapy [91]. EAD1 is a synthesized analogue of hydroxychloroquine (HCQ) that is found to make KRAS-mutant PDAC more sensitive to radiotherapy [92]. A study reported that cetuximab (CTX)-conjugated maleimide-polyethylene glycol-chlorin e6 (CMPC) is an immune-stimulating antibody-photosensitizer conjugate, and it can be used for KRAS-mutant PDAC, the mechanism of which is Fc-mediated dendritic cell phagocytosis and immunogenic cell death triggered by light [93]. Silvestrol (an eIF4A inhibitor) is reported to inhibit the overexpression of ARF6 and MYC driven by KRAS mutations, thus improving the efficacy of immunotherapy for PDAC [94].

In addition, Prochlorperazine (PCZ) is an antipsychotic drug reported to reverse the resistance of KRAS-mutant NSCLC cells to radiotherapy [95]. MicroRNA-16 is also found to restore the sensitivity of TKI therapy, with better performance than MEK inhibitors in KRAS-mutant NSCLC [96]. Melatonin is found to regulate the immunosuppressive tumor microenvironment (TME) by inhibiting the YAP/PD-L1 axis, and it may be a novel therapy for KRAS-mutant NSCLC [97]. Luteolin and its derivative apigenin are reported to significantly suppress the expression of PD-L1 induced by IFN-γ, leading to the better anti-tumor activity of KRAS-mutant lung cancer, and a synergistic effect combined with PD-1 ICIs [98]. Statins can provoke the CD8+ T-cell immune response to KRAS-mutant tumors, and increase the sensitivity to PD-1 ICIs when combined with oxaliplatin [99]. More emerging therapies such as T-cell vaccines and adoptive T-cell therapy (ATC) are also under research [100]. Recent clinical trials of other unconventional therapies are presented in Table 4.

## 5. Advances in Drug Resistance and Oncological Mechanisms of KRAS-Mutant Cancers

Despite the significant innovation and clinical benefits in the treatment strategies for KRAS-mutant patients, drug resistance is still inevitable regardless of the treatment strategies. Therefore, research on drug resistance mechanisms and oncogenic mechanisms is urgently needed to overcome existing drug resistance and to improve the outcome of patients with KRAS mutations. This section will systematically discuss the latest research progress on drug resistance mechanisms and oncogenic mechanisms in patients with KRAS mutations to different therapies.

### 5.1. Drug Resistance of KRAS Inhibitors

The efficacy of KRAS inhibitors for treating patients with KRAS mutations varies. The mechanisms of resistance against KRAS inhibitors mainly include primary resistance and acquired resistance after receiving treatment. This section will elaborate on each in detail.

#### 5.1.1. Primary Resistance

A low reliance on the KRAS pathway is the reason for primary drug resistance. Studies of KRAS-mutated cell lines in lung and pancreatic cancers have revealed that different cell lines have different degrees of dependence on KRAS [101]. A study of PDAC cell lines found that a subset of cells is not dependent on KRAS mutations, but instead is highly dependent on the PI3K-mediated MAPK pathway [102]. Studies have reported that the PI3K pathway can be activated by signalings other than KRAS [103], and that amplification and the overexpression of the transcriptional coactivator Yap1 can also drive KRAS-independent PDAC tumor maintenance [104]. The deubiquitin USP21 was also found to drive resistance to KRAS-targeted therapy in PDAC [105]. Alterations in the tumor immune microenvironment can also lead to primary drug resistance, such as the finding that tumor-associated macrophages can be recruited due to the effect of HDAC5 on chemokine, mediating SMAD4-dependent and KRAS-independent PDAC tumor growth [106].

#### 5.1.2. Acquired Resistance

As the treatment population continues to expand, patients using KRAS inhibitors have developed resistance to the inhibitors, limiting their efficacy [1]. This may be explained with genetic alterations in nucleotide exchange function, adaptive mechanisms in downstream pathways, or newly emerged KRAS G12C mutations [107]. 

Mutations could disrupt covalent or potentially non-covalent drug binding [108]. For instance, clinical studies have reported on the development of the KRAS Y96D mutation in patients resistant to MRTX849, which affects the Switch-II pocket and can lead to resistance to all current KRAS G12C inhibitors [109]. Different KRAS secondary variants also cause resistance to different drugs, as it has been found that the G13D, R68M, A59S, and A59T mutations are resistant to AMG510 but sensitive to MRTX849, while the Q99L mutation has the opposite effect [110].

The negative feedback of upstream signals (such as EGFR) or downstream mediators (such as MEK) has also been demonstrated to attenuate the efficacy of KRAS inhibitors [111,112]. It was found that KRAS G12C inhibition is followed by the rapid adaptive RAS pathway feedback reactivation, driven by the RTK-mediated activation of wild-type RAS, which could not be inhibited by specific G12C inhibitors, requiring the combined inhibition of SHP2 to maintain its efficacy [113]. The EGFR and aurora kinase signaling pathways have also been found to reactivate KRAS through feedback regulation and maintain drug resistance in newly generated KRAS mutations [114].

Dysregulating FAK-YAP signaling and fibrosis formation is also found to cause attenuated treatment outcomes [115]. Inducing EMT to enhance PI3K/AKT signaling and MAPK signaling can also induce drug resistance [111,116]. The Transformation of pathology from adenocarcinoma to squamous cell carcinoma occurs in patients of solid tumors with KRAS mutations [108], and genetic alterations in cell cycle regulators is reported, causing cell cycle dysregulation to attenuate the therapeutic response of KRAS inhibitor [117]. MET amplification is found to induce resistance to KRAS inhibitors in NSCLC patients through RAS-dependent and non-dependent pathways, which can be reversed via MET/KRAS G12C dual inhibition [118]. HER2 has also been found to mediate resistance to KRAS inhibitors, which can be overcome by SHP2/KRAS G12C dual inhibition [119]. The mechanisms of acquired drug resistance are diverse and complex, and some studies have found that multiple mechanisms can occur simultaneously in the same patient [108], warranting further exploration.

### 5.2. Drug Resistances of Other Therapies

Apart from targeted therapies, patients with KRAS mutations also shown resistance to immunotherapy and radiotherapy. A possible mechanism is that KRAS mutations may promote an immunosuppressive TME of CRC through the inhibition of IRF2 and the recruitment of myeloid-derived suppressor cells (MDSCs) [120]. Another study showed that KRAS-G12D mutations initiate the primary resistance of immunotherapy in NSCLC by suppressing the PD-L1 level via the P70S6K/PI3K/AKT axis, and reducing CXCL10/CXCL11 levels via the down-regulation of high mobility group protein A2 (HMGA2) level [121]. STK11/LKB1 mutations have been found to cause primary resistance to PD-1/PD-L1 inhibitors in patients with KRAS-mutated lung adenocarcinoma [122]. KRAS mutations may also cause resistance to radiotherapy through the upregulation of NRF2-53BP1-mediated non-homologous end-joining repair [123]. 

### 5.3. Oncological Mechanisms of KRAS-Mutant Cancers

In addition to the development of more targeted agents to reverse existing drug resistance, preclinical studies in many other directions offer the possibility of future therapeutic directions as well. These studies may provide ideas for further improving patient outcomes, and delaying or overcoming drug resistance. We organized the corresponding research progress into two aspects: tumor cell characterization and immune microenvironment remodeling.

#### 5.3.1. Tumor Proliferation, Survival, and Migration

In PDAC and CRC, a study based on the samples of seven PDAC patients found that blocking RAS downstream signaling and epigenetic pathways could synergistically increase the anti-proliferative activity of KRAS mutant PDAC cells [124]. A study found that PDAC cells regulate pH and glycolysis to increase carbonic anhydrase 9 (CA9) expression by stabilizing hypoxia inducible factor (HIF) 1-Alpha (HIF1A) and HIF2A during hypoxia; thus, the disruption of this pathway may slow the growth of PDAC xenograft tumors in mice, becoming a potential target pathway for pancreatic cancer [125]. A study based on public databases found that the inhibition of tensin 4 (TNS4) may be effective in treating patients with cetuximab-refractory CRC, including activated KRAS mutations [126].

In lung cancer, a study found that KRAS mutations could promote the cell growth of lung cancer cells with SLC3A2-NRG1 (S-N) fusion. An elevation of Ras/Raf/MEK/ERK and ERBB/PI3K/Akt/mTOR pathways through disintegrin and metalloproteinase 17 (ADAM17)-mediated neuregulin 1 (NRG1) shedding was observed, indicating potential vulnerability to MEK1/2 and/or ADAM17 inhibitors in patients with concurrent KRAS mutation and S-N fusion [127]. A study based on cell experiments found that the simultaneous inhibition of activated Cdc42-associated kinase (ACK1) and AKT inhibited the growth and migration of KRAS-mutant NSCLC cells, providing the premise for the clinical translation of ACK1 and AKT inhibitors either as monotherapy or with rational combination [128]. GRP78 haploinsufficiency is reported to inhibit KRAS G12D-mediated tumor progression and prolong survival, and it is a potential therapeutic target for KRAS-mutant lung cancer [129]. It has also been found that regenerating family member (REG4) is highly expressed in KRAS-mutant lung adenocarcinoma, and that silencing REG4 can inhibit cancer cell proliferation and genesis, making it a potential therapeutic target [130]. A study finds that the inhibition of phosphoglucomutase 3 (PGM3) reduces the growth of KRAS/LKB1 co-mutant lung tumors in both in vitro and in vivo settings, suggesting the potential for PGM3 targeted therapy in the KRAS-mutant population [131].

There are also some pan-cancer studies focusing on the oncogenic mechanism of KRAS mutations. KRAS-mutant cancer cells can produce exosomes that are enriched in Survivin, to promote cancer cell survival and resistance to therapy [132]. PTPN2 is reported to regulate the activation of KRAS mutations, as well as the proliferation and survival of cancer cells, making it a potential therapeutic target for KRAS-mutant cancers [58].

#### 5.3.2. The Immune Microenvironment

In PDAC and CRC, a study found that resistance to anti-KRAS therapy may be driven by deubiquitinase USP21, which promotes KRAS-independent tumor growth by regulating MARK3-induced macrophagocytosis, and thus, USP21 may serve as a new target for the treatment of pancreatic cancer [105]. A study of 272 patients with metastatic CRC found statistically significant differences in neutrophil/lymphocyte ratio (*p* = 0.042), and systemic inflammation indices (*p* = 0.004) between the KRAS mutant group and the wild-type group [133]. A study based on 17,909 CRC patients found that rare KRAS mutation subtypes such as A59T were correlated with predictive immunotherapy response biomarkers [134]. The inhibition of ERK signaling was found to assist in reducing PD-L1 expression through autophagy in intrahepatic cholangiocarcinoma (iCCA), indicating that ERK-targeted therapy may be combined with anti-PD-(L)1 immunotherapy for KRAS-mutant iCCA [135]. Equipping cetuximab on the surface of NK cells may solve the problem of cetuximab resistance in KRAS-mutant CRC [136]. 

In NSCLC, PD-L1 expression was found to be induced in KRAS G12V-mutant NSCLC and promote immune escape through the transforming growth factor (TGF)-β/EMT signaling pathway [137]. The correlation between low serum deprivation protein response (low SDPR) and immunosuppression in KRAS-mutant NSCLC was found, making low SDPR a possible prognostic factor for worse prognosis in KRAS-mutant NSCLC [56]. Th17 is found to cause resistance to MEK inhibitor combined with PD-L1 inhibitor therapy in lung cancer patients with KRAS/p53 mutations [138]. It was found that TRIM58 was positively correlated with an abundance of M2 macrophages and resting mast cells in KRAS-mutant lung adenocarcinoma, and negatively correlated with an abundance of follicular helper T cells [139]. According to TCGA database, the PD-L1 protein expression level and immune cell infiltration are significantly decreased in the KRAS G12D/TP53 mutant group. Such co-mutation drives immunosuppression and may be a negative predictive biomarker for anti-PD-(L)1 ICIs in patients with lung adenocarcinoma [140].

In the research regarding all solid tumors, the KRAS mutation is reportedly immunogenic for CD4+ T cells and is a potential target for T-cell receptor (TCR)-based immunotherapy [141]. A study found that regulating ROS or inhibiting fibroblasts growth factor receptor 1 (FGFR1) signaling could abrogate the immunosuppression mediated by PD-L1, improving the efficacy of immunotherapy in KRAS-mutant cancers [142].

## 6. Conclusions and Future Perspectives

As mentioned above, immunotherapy is currently used mainly in the real world for patients with KRAS-mutant solid tumors. Clinical studies are gradually phased into target therapies, and more preclinical studies are considering other therapies such as T-cell vaccines. Sotorasib is the only FDA-approved KRAS G12C inhibitor, and has shown satisfying results in real-world studies. Immunotherapy alone or combined with chemotherapy has been proven to be effective in treating patients with KRAS mutations as well.

However, more KRAS inhibitors targeting non-G12C subtypes are yet to be invented, and the acquired resistance of KRAS G12C inhibitors may reduce their efficacy. Co-mutations have the potential of interfering with the binding of KRAS inhibitors and their receptors, which is now the most acknowledged resistance mechanism. More research regarding the restoration of the sensitivity of targeted therapy is needed, and the immune microenvironment of KRAS-mutant tumors is worth exploring. In conclusion, the future of solid tumors with KRAS mutations is promising in terms of developing therapeutic strategies and overcoming drug resistance.

## Figures and Tables

**Figure 1 jcm-12-00709-f001:**
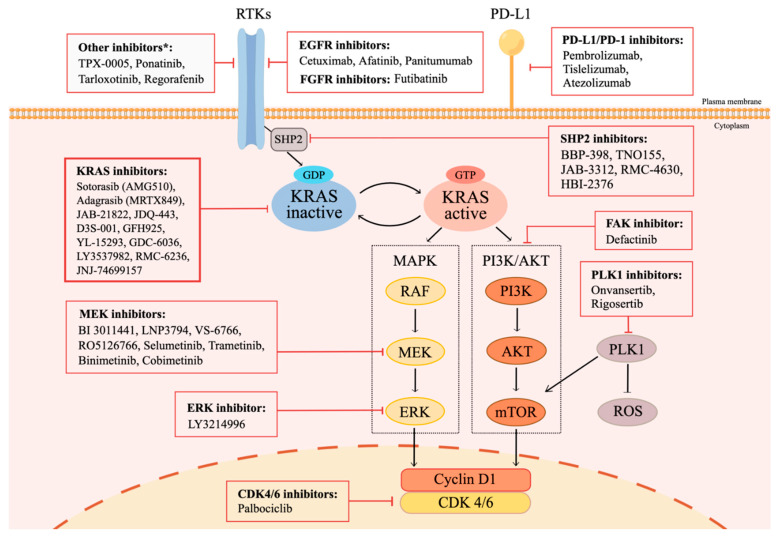
Signaling Pathways Related to KRAS Mutations, and Potential Treatment Strategies. * KRAS switches between a guanosine diphosphate (GDP)-bound inactive state and a guanosine triphosphate (GTP)-bound active state. Normally, KRAS is bound to GDP and remains inactive. Activation through receptor tyrosine kinases (RTKs) leads to the activation of the guanine nucleotide exchange factor (GEF) family, which subsequently triggers the exchange between GDP and GTP. GTP-bound active KRAS transduces downstream signals, including mitogen-activated protein kinase (MAPK) pathway and the phosphoinositide 3-kinase (PI3K) pathway, which are responsible for cell proliferation, cell cycle regulation, cell survival, and cell differentiation. The treatment approaches of KRAS mutant patients include therapies targeting KRAS, and factors involved in the KRAS mutation pathways, such as RTKs, SHP2, PI3K pathway elements, MAPK pathway elements, and CDK4/6. ICIs are also included, considering the potential effect on KRAS mutant patients. This figure was created using Figdraw.

**Table 1 jcm-12-00709-t001:** KRAS mutation rate and subtype proportion in common cancers.

Cancer Type	KRAS Mutation				
	N of Samples	Rate (%)	Top 3 Subtypes (Proportion of All KRAS Mutations, %)		
Pan-cancer	87,606	11.60	G12D (29.19)	G12V (22.97)	G12C (13.43)
Pancreatic adenocarcinoma	990	81.72	G12D (40.20)	G12V (31.96)	G12R (17.10)
Colorectal carcinoma	3853	37.97	G12D (28.04)	G12V (18.50)	G13D (18.10)
Non-small cell lung cancer	4584	21.20	G12C (45.42)	G12V (15.78)	G12D (13.03)

Data acquired from cBioPortal.org. G12: codon 12 encoding glycine; G13: codon 13 encoding glycine.

**Table 2 jcm-12-00709-t002:** Clinical Trials of KRAS G12C-Targeted Therapy on KRAS Mutant Tumors.

Drug Name	Treatment Strategy	Stage	Patient Characteristics	Number of Patients	Initiation Year	NCT Number
Sotorasib (AMG 510)	Monotherapy	Phase 2	Advanced NSCLC with KRAS G12C mutations	116	2021	NCT04625647
	Monotherapy	Phase 1–2	Advanced solid tumors with KRAS G12C mutations	793	2018	NCT03600883
	Monotherapy	Phase 2	Stage IV NSCLC with KRAS G12C mutations without prior treatment	170	2022	NCT04933695
	Monotherapy	Phase 2	Stage Ib-IIIA resectable NSCLC with KRAS G12C mutations	25	2022	NCT05400577
	Monotherapy	Phase 2	Stage III unresectable NSCLC with KRAS G12C mutations	43	2022	NCT05398094
	Monotherapy	Phase 1	Advanced solid tumors with KRAS G12C mutations	12	2020	NCT04380753
	Monotherapy	Phase 2	Stage III unresectable NSCLC with KRAS G12C mutations	43	2022	NCT05398094
	Monotherapy (VS Docetaxel)	Phase 3	Advanced NSCLC with KRAS G12C mutations	345	2020	NCT04303780
	Combined with Tarloxotinib (pan-ERBB inhibitor)	Phase 1–2	Advanced NSCLC with KRAS G12C mutations	30	2022	NCT05313009
	Combined with BBP-398 (SHP2 inhibitor)	Phase 1	Advanced solid tumors with KRAS G12C mutations	85	2022	NCT05480865
	Combined with VS-6766 (RAF/MEK inhibitor)	Phase 1–2	Advanced NSCLC with KRAS G12C mutations	53	2022	NCT05074810
	Combined with targeted therapy, chemotherapy, or immunotherapy	Phase 1–2	Advanced Solid tumors with KRAS G12C mutations	1054	2019	NCT04185883
	Combined targeted therapy, chemotherapy, or immunotherapy	Phase 1–2	Advanced solid tumors with KRAS G12C mutations	1054	2019	NCT04185883
	Combined with Panitumumab (anti-EGFR mAb) vs. Trifluridine and Tipiracil (chemotherapy) + Regorafenib (multi-kinase inhibitor *)	Phase 3	Advanced CRC with KRAS G12C mutations	153	2022	NCT05198934
	Combined with MVASI (antiangiogenic drug)	Phase 1–2	Advanced NSCLC with KRAS G12C mutations and Brain metastasis	43	2022	NCT05180422
	Combined with Cisplatin or Carboplatin and Pemetrexed (chemotherapy)	Phase 2	Stage IIA-IIIB resectable non-squamous NSCLC with KRAS G12C mutations	27	2022	NCT05118854
Adagrasib (MRTX849)	MRTX monotherapy or combined with Pembrolizumab (anti-PD-1 ICI)/Cetuximab (anti-EGFR IgG1 mAb)/Afatinib (EGFR TKI)	Phase 1–2	Advanced or metastatic cancer with KRAS G12C mutations	740	2019	NCT03785249
	Monotherapy	Phase 2	Advanced or metastatic NSCLC with KRAS G12C mutations	116	2022	NCT03785249
	MRTX849 monotherapy or combined with Pembrolizumab (anti-PD-1 ICI)	Phase 2	Advanced or metastatic NSCLC with KRAS G12C mutations	250	2020	NCT04613596
	Monotherapy vs. Docetaxel (chemotherapy)	Phase 3	Advanced or metastatic NSCLC	340	2021	NCT04685135
	MRTX849 combined with Cetuximab (anti- EGFR IgG1 mAb) vs. mFOLFOX6 and FOLFIRI (chemotherapy)	Phase 3	Advanced CRC with KRAS G12C mutations	420	2021	NCT04793958
	Combined with VS-6766 (RAF-MEK inhibitor)	Phase 1–2	Advanced NSCLC with KRAS G12C mutations	85	2022	NCT05375994
JAB-21822	Monotherapy	Phase 1–2	Advanced solid tumors with KRAS G12C mutations	144	2021	NCT05009329
	Monotherapy or combined with Cetuximab (anti-EGFR IgG1 mAb)	Phase 1–2	Advanced solid tumors with KRAS G12C mutations	100	2021	NCT05002270
	Combined with Cetuximab (anti-EGFR IgG1 mAb)	Phase 1–2	Advanced solid tumors with KRAS G12C mutations	62	2022	NCT05194995
	Combined with JAB-3312 (SHP2 inhibitor)	Phase 1–2	Advanced solid tumors with KRAS G12C mutations	124	2022	NCT05288205
JDQ443	Monotherapy	Phase 3	Advanced NSCLC with KRAS G12C mutations	360	2022	NCT05132075
	Monotherapy or combined with TNO155 (SHP2 inhibitor) or tislelizumab (anti-PD-1 ICI) or TNO155 + tislelizumab	Phase 1–2	Advanced solid tumors with KRAS G12C mutations	425	2021	NCT04699188
D3S-001	Monotherapy	Phase 1	Advanced solid tumors with KRAS G12C mutations	98	2022	NCT05410145
GFH925	Monotherapy	Phase 1–2	Advanced solid tumors with KRAS G12C mutations	128	2021	NCT05005234
YL-15293	Monotherapy	Phase 1–2	Advanced solid tumors with KRAS G12C mutations	55	2021	NCT05119933
JNJ-74699157	Monotherapy	Phase 1	Advanced solid tumors with KRAS G12C mutations	10	2019	NCT04006301
GDC-6036	Monotherapy or combined with chemotherapy, immunotherapy, etc.	Phase 1	Advanced solid tumors with KRAS G12C mutations	498	2020	NCT04449874
LY3537982	Monotherapy or combined with targeted therapy, immunotherapy, etc.	Phase 1	Advanced solid tumors with KRAS G12C mutations	360	2021	NCT04956640
RMC-6236	Monotherapy (KRAS G12X inhibitor)	Phase 1	Advanced solid tumors with KRAS mutations	141	2022	NCT05379985

* NSCLC: non-small cell lung cancer; CRC: colorectal cancer; SHP2: Src homology2 (SH2) domain-containing protein tyrosine phosphatase (PTPase); EGFR: epidermal growth factor receptor; MEK: mitogen-activated extracellular signal-regulated kinase; IgG1: immunoglobulin G1; mAb: monoclonal antibody; ICI: immune checkpoint inhibitor; TKI: Tyrosine kinase inhibitor.

**Table 3 jcm-12-00709-t003:** Clinical Trials of Non-KRAS-Targeted Therapies on KRAS Mutant Tumors.

Drug Definition	Drug Name	Treatment Strategy	Stage	Patient Characteristics	Number of Patients	Initiation Year	NCT Number
MEK inhibitors	BI 3011441	Monotherapy	Phase 1	Advanced, unresectable or metastatic refractory solid tumors with NRAS/KRAS mutations	15	2021	NCT04742556
	LNP3794	Monotherapy	Phase 1	Advanced or metastatic refractory solid tumors with NRAS/KRAS mutations	15	2020	NCT05187858
	RO5126766	Monotherapy	Phase 1	Advanced NSCLC with KRAS mutations	15	2018	NCT03681483
	Trametinib	Combined with Pembrolizumab (anti-PD-1 ICI)	Phase 1	Stage IV NSCLC with KRAS mutations	15	2018	NCT03299088
	Ponatinib; Trametinib	Combined with Ponatinib (BCR-ABL TKI)	Phase 1–2	Advanced NSCLC with KRAS mutations	12	2018	NCT03704688
	TPX-0005; Trametinib	Combined with TPX-0005 (ROS1/TRK/ALK inhibitor)	Phase 1–2	Advanced or metastatic solid tumors with KRAS mutations	74	2021	NCT05071183
	Trametinib; Anlotinib	Combined with Anlotinib (antiangiogenic drug)	Phase 1	Advanced NSCLC with KRAS mutations	30	2021	NCT04967079
	Trametinib; Hydroxychloroquine	Combined with Hydroxychloroquine (chemotherapy)	Phase 2	Refractory BTC with KRAS mutations	30	2022	NCT04566133
MEK inhibitors	Binimetinib; Hydroxychloroquine	Combined with Hydroxychloroquine (chemotherapy)	Phase 1	Advanced PDAC with KRAS mutations	39	2019	NCT04132505
	Binimetinib; Hydroxychloroquine	Combined with Hydroxychloroquine (chemotherapy)	Phase 2	Advanced NSCLC with KRAS mutations	29	2021	NCT04735068
	Binimetinib; Hydroxychloroquine	Combined with Hydroxychloroquine (chemotherapy)	Phase 2	Advanced NSCLC with KRAS mutations	29	2021	NCT04735068
	Binimetinib; Futibatinib	Combined with Futibatinib (FGFR 1–4 inhibitor)	Phase 1–2	Advanced or Metastatic Solid Tumors with KRAS mutations	36	2021	NCT04965818
	Binimetinib; Pemetrexed and Cisplatin	Combined with Pemetrexed and Cisplatin (chemotherapy)	Phase 1	Advanced NSCLC with KRAS mutations	18	2017	NCT02964689
	Binimetinib; Palbociclib; Trifluridine and Tipiracil Hydrochloride	Binimetinib + Palbociclib (CDK4/6 Inhibitor) vs. Trifluridine and Tipiracil Hydrochloride (chemotherapy)	Phase 2	Advanced CRC with KRAS or NRAS mutations	101	2019	NCT03981614
	Binimetinib; Palbociclib	Combined with Palbociclib (CDK4/6 inhibitor)	Phase 1–2	Advanced NSCLC with KRAS mutations	72	2017	NCT03170206
	Cobimetinib; Hydroxychloroquine; Atezolizumab	Combined with Hydroxychloroquine (chemotherapy) and Atezolizumab (anti-PD-L1 ICI)	Phase 1–2	Advanced solid tumors with KRAS mutations	175	2020	NCT04214418
MEK inhibitors	VS-6766; Defactinib	Monotherapy vs, combination therapy of VS-6766 and Defactinib (FAK inhibitor)	Phase 2	Recurrent NSCLC with KRAS and BRAF mutations	100	2020	NCT04620330
SHP2 inhibitors	HBI-2376	Monotherapy	Phase 1	Advanced malignant solid tumors with KRAS or EGFR mutations	42	2021	NCT05163028
	RMC-4630; LY3214996	Combined with LY3214996 (ERK1/2 inhibitor)	Phase 1	Metastatic solid tumors with KRAS mutations	55	2022	NCT04916236
	BBP-398 with nivolumab	Combined with nivolumab (anti-PD-1 ICI)	Phase 1	Advanced NSCLC with KRAS mutations	45	2022	NCT05375084
Multi-targeting kinase inhibitor	Regorafenib; Methotrexate	Combined with Methotrexate (chemotherapy)	Phase 2	Recurrent or metastatic NSCLC with KRAS mutations	18	2018	NCT03520842
PLK1 inhibitors	Onvansertib; Bevacizumab; FOLFIRI	Combined with Bevacizumab (antiangiogenic drug) and (FOLFIRI: chemotherapy)	Phase 1–2	Metastatic CRC with KRAS mutations	100	2019	NCT03829410
	Rigosertib; Nivolumab	Combined with Nivolumab (anti-PD-1 antibody)	Phase 1–2	Stage IV NSCLC with KRAS mutations	20	2020	NCT04263090

NSCLC: non-small cell lung cancer; CRC: colorectal cancer; BTC: biliary tract cancer; PDAC: pancreatic ductal adenocarcinoma; MEK: mitogen-activated extracellular signal-regulated kinase; TRK: tropomyosin receptor kinase; ALK: anaplastic lymphoma kinase; FGFR: fibroblasts growth factor receptor; FAK: focal adhesion kinase; ERK: extracellular regulated protein kinases; CDK4/6: cyclin-dependent kinases 4/6; ICI: immune checkpoint inhibitor; TKI: Tyrosine kinase inhibitor.

**Table 4 jcm-12-00709-t004:** Clinical Trials of Other Unconventional Therapies on KRAS Mutant Tumors.

Drug Name	Drug Definition	Treatment Strategy	Stage	Patient Characteristics	Number of Patients	Initiation Year	NCT Number
TVB-2640	Fatty acid synthase (FASN) inhibitor	Monotherapy	Phase 2	Metastatic or advanced NSCLC with KRAS mutations	12	2019	NCT03808558
ELI-002	KRAS therapeutic vaccine	Monotherapy	Phase 1	Solid tumors with KRAS mutations	18	2021	NCT04853017
REOLYSIN	Reovirus	Combined with FOLFIRI and Bevacizumab (chemotherapy and antiangiogenic drug)	Phase 1	Metastatic CRC with KRAS mutations	36	2010	NCT01274624
Mutant KRAS G12V-specific TCR transduced autologous T cells	Mutant KRAS G12V-specific TCR transduced autologous T cells	Chemotherapy prior to combination therapy of Mutant KRAS G12V-specific TCR transduced autologous T cells and Anti-PD-1 monoclonal antibody	Phase 1–2	Advanced PDAC with KRAS G12V mutations	30	2021	NCT04146298

NSCLC: non-small cell lung cancer; CRC: colorectal cancer; PDAC: pancreatic ductal adenocarcinoma; FASN: fatty acid synthase; TCR: T-cell receptors.

## Data Availability

Not applicable.

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
