# Peer review of "KRAS Mutations in Solid Tumors: Characteristics, Current Therapeutic Strategy, and Potential Treatment Exploration"

_jcm, 2023, doi:10.3390/jcm12020709_

Round 1

Reviewer 1 Report

The authors have written an extensive overview of KRAS in cancer. My knowlegde is limited to clinical part of NSCLC so i have limited my view to mainly  that part. 

Introduction: 

More emphasis is needed about treatment including Adagrasib  https://www.nejm.org/doi/full/10.1056/NEJMoa2204619 why 

Prognosis: Although conflicting evidence is available of which a selection has been proparly made the consensus is that KRAS had no impact on the prognosis in NSCLC. 

Immunotherapy in KRAS mutated cancers

The section about combinations of chemo immunotherapy in NSCLC suggests that chemo immuno is better than immuno only, however it must be added that the PDL-1 status drives the data what is better rather then the KRAS status. 

Kras targetd therapy

This might have been written prior the the NEJm adagrasib paper, however now that one has been published it cannot be remain unnamed. This sections needs mayor revision the include the data and lessons learned from this study, 

Furthermore it should be disucced where the optimal place for KRAS inhibitors will be, first or second line? Alone or in combination with chemo/IO?

Will it remain in NSCLC or will it come to other cancers? 

Next it is important to explain better in the paper why after so many decades of being un drugable now we finally seem to have drugs for G12C, but not for other KRAS. 

Author Response

Reviewer 1

Comment 1: Introduction: More emphasis is needed about treatment including Adagrasib  https://www.nejm.org/doi/full/10.1056/NEJMoa2204619

Response and changes: Thank you for your suggestion. We found this comment extremely helpful and have revised accordingly, shown in line 38-39, 305-306 and Table 2.

Comment 2: Prognosis: Although conflicting evidence is available of which a selection has been properly made the consensus is that KRAS had no impact on the prognosis in NSCLC.

Response and changes: Thank you for your constructive suggestion. We concur with this comment and have added the consensus in line 162-163.

Comment 3: Immunotherapy in KRAS mutated cancers: The section about combinations of chemo immunotherapy in NSCLC suggests that chemo immuno is better than immuno only, however it must be added that the PDL-1 status drives the data what is better rather than the KRAS status.

Response and changes: Thank you for your constructive comment. We have rephrased and emphasized such findings in line 291-294.

Comment 4: Kras-targeted therapy: This might have been written prior the NEJm adagrasib paper, however now that one has been published it cannot remain unnamed. This section needs major revision that include the data and lessons learned from this study. Furthermore, it should be discussed where the optimal place for KRAS inhibitors will be, first or second line? Alone or in combination with chemo/IO? Will it remain in NSCLC or will it come to other cancers?

Response and changes: We are appreciative of your valuable comment. We found this comment insightful and have made accordingly alterations in line 38-39, 305-306 and Table 2 of updated Adagrasib findings, and line 314-317 of the optimal place for KRAS inhibitors.

Comment 5: Next it is important to explain better in the paper why after so many decades of being undruggable now we finally seem to have drugs for G12C, but not for other KRAS.

Response and changes: Thank you for your precious comment. We have rephrased and illustrated the difficulty of developing KRAS inhibitors in line 298-301 and the reason why we have only seen results of KRAS G12C instead of other subtypes in line 311-314.

Reviewer 2 Report

This manuscript provided a systematic overview of the KRAS pathway in solid tumors, and presented a landscape of KRAS mutation in cancer patients. In addition, this text focused on the role of KRAS mutations in PDACCRC, and NSCLC (the top three malignancies with KRAS mutations)described the prognostic effect it posed on different therapies. More meaningful, this review detailed advanced therapeutic strategy, as well as cutting-edge research on mechanisms of drug resistance, tumor development and immune microenvironment. Overall, this is a well-written and clearly structured manuscript, but several concerns remain:

1. In 3.1, the frequency and types of KRAS mutations in different cancers was suggested to be more clearly shown in a figure or table.

2. It has been mentioned that KRAS co-mutations can affect the clinicopathological features and prognosis of cancer patients. Although the impact of KRAS co-alterations on prognosis has been elaborated in 3.3.3, it lacks a detailed description of features such as the type and frequency of co-mutations.

3. In this new era of targeting KRAS mutations, it is important to understand and overcome the drug resistance mechanisms. Although some resistance mechanisms are mentioned in lines 380-388, they seem to be incomplete and can be further described in detail by dividing them into primary and acquired resistance.

4. Some abbreviations require the full name when they first appear.

5. Statistical P-values need italics.

Author Response

Reviewer Comments, Author Responses and Manuscript Changes

Reviewer 2

Comment 1: In 3.1, the frequency and types of KRAS mutations in different cancers was suggested to be more clearly shown in a figure or table.

Response and changes: Thank you for such an enlightening suggestion. For better comprehension, we have added a new Table now as Table.1, and have updated related data in section 3.1 shown in line 67-91.

Comment 2: It has been mentioned that KRAS co-mutations can affect the clinicopathological features and prognosis of cancer patients. Although the impact of KRAS co-alterations on prognosis has been elaborated in 3.3.3, it lacks a detailed description of features such as the type and frequency of co-mutations.

Response and changes: Thank you for your constructive comment. We have added a detailed description of co-mutations of KRAS mutations in line 181-189 as suggested.

Comment 3: In this new era of targeting KRAS mutations, it is important to understand and overcome the drug resistance mechanisms. Although some resistance mechanisms are mentioned in lines 380-388, they seem to be incomplete and can be further described in detail by dividing them into primary and acquired resistance.

Response and changes: Thank you for your insightful ideas. We have rephrased and reorganized contents regarding the drug mechanisms of KRAS-related therapies in section 5, line 406-493 by dividing primary and acquired drug resistance.

Comment 4: Some abbreviations require the full name when they first appear.

Response and changes: Thank you for your rigorous suggestion. We have carefully checked the entire manuscript to fix this issue.

Comment 5: Statistical P-values need italics.

Response and changes: Thank you for pointing out these errors. We have gone through the entire manuscript carefully and corrected related fonts.

Round 2

Reviewer 1 Report

Thank you, the authors have incorporated my comments and improved the paper substantially